# Subcellular Localization of uc.8+ as a Prognostic Biomarker in Bladder Cancer Tissue

**DOI:** 10.3390/cancers13040681

**Published:** 2021-02-08

**Authors:** Sara Terreri, Sara Mancinelli, Matteo Ferro, Maria Concetta Vitale, Sisto Perdonà, Luigi Castaldo, Vincenzo Gigantino, Vincenzo Mercadante, Rossella De Cecio, Gabriella Aquino, Marco Montella, Claudia Angelini, Eugenio Del Prete, Marianna Aprile, Angelo Ciaramella, Giovanna L. Liguori, Valerio Costa, George A. Calin, Evelina La Civita, Daniela Terracciano, Ferdinando Febbraio, Amelia Cimmino

**Affiliations:** 1Institute of Genetics and Biophysics “A. Buzzati Traverso”, National Research Council (CNR), 80131 Naples, Italy; sara.terreri@opbg.net (S.T.); sara.mancinelli@humanitasresearch.it (S.M.); vincenzo.mercadante@igb.cnr.it (V.M.); marianna.aprile@igb.cnr.it (M.A.); giovanna.liguori@igb.cnr.it (G.L.L.); valerio.costa@igb.cnr.it (V.C.); 2Immunology Research Area, B-cell development Unit, Children Hospital Bambino Gesù, 00146 Rome, Italy; 3Department of Biomedical Sciences, Humanitas University, 20090 Pieve Emanuele, Milan, Italy; 4IRCCS Humanitas Research Hospital, 20089 Rozzano, Milan, Italy; 5Division of Urology, European Institute of Oncology IRCCS, 20141 Milan, Italy; matteo.ferro@ieo.it; 6Department of Science and Technology, University of Naples Parthenope, 80143 Naples, Italy; mariaconcetta.vitale001@studenti.uniparthenope.it (M.C.V.); angelo.ciaramella@uniparthenope.it (A.C.); 7Uro-Gynecological Department, Istituto Nazionale per lo Studio e la Cura dei Tumori, Fondazione “G. Pascale”-IRCCS, 80131 Naples, Italy; s.perdona@istitutotumori.na.it (S.P.); luigi.castaldo@istitutotumori.na.it (L.C.); 8Pathology Unit, Istituto Nazionale per lo Studio e la Cura dei Tumori, Fondazione “G. Pascale”-IRCCS, 80131 Naples, Italy; gigantino.vincenzo@gmail.com (V.G.); r.dececio@istitutotumori.na.it (R.D.C.); g.aquino@istitutotumori.na.it (G.A.); 9Section of Pathology, Department of Physical and Mental Health and Preventive Medicine, Università degli studi della Campania “Luigi Vanvitelli”, 80131 Naples, Italy; marco.montella@unicampania.it; 10Institute for Applied Mathematics “Mauro Picone”, National Research Council (CNR), 80131 Naples, Italy; c.angelini@iac.cnr.it (C.A.); e.delprete@na.iac.cnr.it (E.D.P.); 11Translational Molecular Pathology Department and Leukemia Department, University of Texas MD Anderson Cancer Center, Houston, TX 77030, USA; gcalin@mdanderson.org; 12Department of Translational Medical Sciences, University of Naples “Federico II”, 80131 Naples, Italy; e.lacivita@studenti.unina.it; 13Institute of Biochemistry and Cell Biology, National Research Council (CNR), 80131 Naples, Italy

**Keywords:** bladder cancer, prognostic biomarker, ultraconserved region, transcribed-ultraconserved region, long noncoding RNA

## Abstract

**Simple Summary:**

DNA regions having high sequence similarity among human, rat and mouse genomes are defined as Ultraconserved Regions. Non-coding RNA transcripts originating by these regions may play relevant roles in the onset and progression of multiple cancer types. We recently found that ultra-conserved-transcript-8+ (uc.8+) levels correlate with the grading and staging of bladder cancer. The aim of this study is to systematically evaluate the expression of ultra-conserved-transcript-8+ (uc.8+) in biopsies and assess its intracellular localization. Furthermore, we aimed to correlate uc.8+ levels with clinical parameters and patient survival. Our analysis indicates that uc.8+ can localize both in the cytoplasm and nucleus of bladder cells at early stages of tumorigenesis, while in tumors at advanced stages, uc.8+ has a prevalent cytoplasmic localization. These data provide relevant information about uc.8+ localization as a hallmark of tumor stage. Finally, using advanced computer-based techniques, we predicted the binding of uc.8+ to RNA-binding proteins. Our study overall suggests that uc.8+ localization can be used as a prognostic biomarker for bladder cancer.

**Abstract:**

Non-coding RNA transcripts originating from Ultraconserved Regions (UCRs) have tissue-specific expression and play relevant roles in the pathophysiology of multiple cancer types. Among them, we recently identified and characterized the ultra-conserved-transcript-8+ (uc.8+), whose levels correlate with grading and staging of bladder cancer. Here, to validate uc.8+ as a potential biomarker in bladder cancer, we assessed its expression and subcellular localization by using tissue microarray on 73 human bladder cancer specimens. We quantified uc.8+ by in-situ hybridization and correlated its expression levels with clinical characteristics and patient survival. The analysis of subcellular localization indicated the simultaneous presence of uc.8+ in the cytoplasm and nucleus of cells from the Low-Grade group, whereas a prevalent cytoplasmic localization was observed in samples from the High-Grade group, supporting the hypothesis of uc.8+ nuclear-to-cytoplasmic translocation in most malignant tumor forms. Moreover, analysis of uc.8+ expression and subcellular localization in tumor-surrounding stroma revealed a marked down-regulation of uc.8+ levels compared to the paired (adjacent) tumor region. Finally, deep machine-learning approaches identified nucleotide sequences associated with uc.8+ localization in nucleus and/or cytoplasm, allowing to predict possible RNA binding proteins associated with uc.8+, recognizing also sequences involved in mRNA cytoplasm-translocation. Our model suggests uc.8+ subcellular localization as a potential prognostic biomarker for bladder cancer.

## 1. Introduction

Bladder cancer (BlCa) is among the most common malignant tumors of the urological system, ranking ninth in worldwide cancer incidence and fourth in men in the USA and is one of the most expensive malignancies, in terms of treatment costs worldwide [1]. The most common form is non-muscle-invasive bladder cancer (NMIBC), which accounts for 75% of newly diagnosed BlCa, while the muscle-invasive form (MIBC) accounts for the remaining part [2,3]. A total of two-thirds of the cases are diagnosed around 65 years or over, and about 23% of diagnosed patients (pts) are 80 years old and over. Biomarkers are necessary to transform bladder cancer management and usher in the age of personalized medicine. Molecular markers for early detection could be the most effective method of BlCa prevention, however the molecular mechanisms underlying BlCa progression are still not well characterized. Numerous publications have suggested that lncRNAs, such as UCA-1, MALAT1, PANDAR and so on, play key roles in development and progression of bladder cancer [4]. Nevertheless, their clinical use is still limited and none is yet validated or widely used in the clinical practice [5].

Ultraconserved regions (UCRs) are approximately 480 sequences in the human genome, with a 100% identity with orthologous sequences in rats and mice, pointing out that they went through a very strong negative selection for 300–400 million years. For these reasons, these regions are called “ultraconserved” [6]. UCRs can be transcribed (T-UCRs) and their non-coding RNA transcripts have different expression profiles and participate with functional roles in the pathophysiology of multiple cancers [6]. In many cases, the function of these intriguing family of long ncRNAs (lncRNAs) is still to be explained; some lncRNAs are likely involved in splicing [7], others map next to transcriptional regulators or developmental genes, suggesting a role related to them [8], others are probably related to cell proliferation, since they show copy number abnormalities in cancer tissues [9]. In literature, molecular details in terms of RNA size and sequence, or mechanisms of action have been described, only for 19 T-UCRs (3.95%) and the subcellular localization has been studied only for 13 (2.7%), highlighting the urgency of better understanding their molecular features [10]. Similar to proteins, lncRNAs exhibit diverse subcellular levels of accumulation extending from predominant nuclear foci to almost exclusively cytoplasmic localization, where they exert distinct regulatory effects [11,12]. Thus, the subcellular localization of lncRNAs plays an important role for their biological function. Nevertheless, many lncRNAs are discretely distributed in different cellular compartments, and the related biological significance still remains largely unclear. In BlCa tissues, upregulation of uc.8+ was inversely related to grade and stage [13], suggesting an early alteration of uc.8+ expression in acquisition of the malignant phenotype. These data support the use of uc.8+ as a possible tissue biomarker in BlCa. In the present work, we focused on the expression of a newly identified lncRNA containing the transcribed ultraconserved region 8 (uc.8+) in samples from pts affected by BlCa. The uc.8+ is transcribed from the intron 1 of CASZ1 gene, encoding for a zinc-finger transcription factor with tumor-suppressing properties. The uc.8+ is overexpressed in BlCa where it behaves as an oncogene [14].

In this study, we further evaluated by RNA in situ hybridization the level and distribution of the uc.8+ in a panel of 73 human BlCa specimens. Moreover, we adopted DeepLncRNA (Deep Learning of Nuclear Classification of long non-coding RNAs) [15], a novel Deep Learning approach developed for predicting lncRNA subcellular localization directly from lncRNA nucleotide sequences. Finally, we investigated whether uc.8+ expression and localization could be associated with BlCa phenotype. Overall, our data point to uc.8+ localization as a prognostic biomarker for bladder cancer.

## 2. Results

### 2.1. Association of uc.8+ Expression with Its Clinic-Pathologic Features and Survival of BlCa Patients

Our previous qPCR-based expression analysis [14] carried out on RNA isolated from *n* = 40 entire BlCa biopsies, revealed that uc.8+ levels correlate both with grading (i.e., cell differentiation) and staging (i.e., tumor invasiveness). However, due to high tumor heterogeneity and the peculiar cell-specific expression of the lncRNAs, we sought to confirm our previous findings by using different techniques, such as ISH. Therefore, we extended the analysis to an independent and larger cohort (*n* = 73) of well-characterized BlCa pts (Table 1).

In summary, about two-thirds of pts have an age ≥ 60 years and, in agreement with the incidence of BlCa disease, there is a prevalence of male pts [2]. The cohort’s pts were divided by tumor grading in two groups: High Grade (HG), consisting of 50 pts with grade 2 and 3 (G2–G3), and Low Grade (LG), consisting of 23 pts with grade 1 (G1). The uc.8+ expression pattern in this cohort of pts was detected by using the ISH analysis (Figure 1A), coloring in blue the digoxigenin-labeled RNA antisense uc.8+ probe by NBT-BCIP, we used normal surrounding urothelium (NSU) of patients as control (Appendix A). 

We detected uc.8+ expression in all analyzed samples. All tissue samples were scored and categorized computing uc.8+ expression level by using the Mean Gray Intensity (MGI) score, taking advantage of ImageJ software as described in materials and methods. According to the mean value (mean = 35.03) of uc.8+ expression in BlCa tissues determined by signal intensity score (SI), the cohort of 73 pts was categorized in two groups showing “High Intensity-score” (HI-s) with SI values ≥ 35 (*n* = 27 pts representing 37% of the cohort) and “Low Intensity-score” (LI-s) with SI values < 35 (*n* = 46 pts representing 63% of the cohort) (Table 1). Univariate analyses showed that increased expression of uc.8+ associates with the prognosis of BlCa pts for 5-year overall survival rates (LI-s/HI-s hazard ratio 3.8, 95% CI; 1.25–11.6; HI-s/Li-s hazard ratio 0.26, 95% CI; 0.09–0.8; *p* < 0.02; r-value = −0.997; 95% CI; −0.9986–−0.9946; Figure 1B). The median overall survival (OS) in the study cohorts was followed for 60.87 months. The mean survival times for BlCa pts were 42.8 and 31.4 months for uc.8+ HI-s and LI-s, respectively. Kaplan–Meier curve shows that low level of uc.8+ is associated with a worst overall survival in BlCa pts dataset. In fact, the prognosis of BlCa pts with LI-s of uc.8+ was significantly poorer than that of BlCa pts with HI-s of uc.8+ (HI-s/LI-s hazard ratio 0.24, 95% CI; 0.07–0.8; LI-s/HI-s hazard ratio 4.2, 95% CI; 1.3–13.8; *p* < 0.05) (Figure 1B). The correlation between uc.8+ SI and clinicopathological findings revealed that uc.8+ expression is strongly related to the pathological grade (*p* = 0.002198, median value LG = 49.7, median value HG = 30.3) (Figure 1C,D) and to the pathological stage (*p* = 0.01373, median value Ta = 53.3, median value T1–2 = 30.4, median value T3–4 = 30.8) (pTNM classification) (Figure 1E,F). In particular, the uc.8+ expression is significantly increased (*p*-value < 0.0001) in LG compared to HG pts, and is significantly higher in Ta clinical stage. No association with age, sex (Appendix A) or neo/adjuvant chemotherapy was observed (all *p*-values < 0.05) (Appendix A) (Table 1). In addition, by using the multiple logistic regression model on *n* = 73 patients, we found a significant association between uc.8+ intensity and the tumor grade (*p* value = 0.00365) after correcting for age and sex. The Odd-Ratio (OR = 0.94, CI = (0.90–0.98)) shows that the worse grade (HG) is associated with a decreased uc.8+ intensity.

### 2.2. In Silico Analysis Indicates That uc.8+ Has Both Nuclear and Cytoplasmic Binding Partners

To predict the subcellular localization of uc.8+ transcript we used a deep learning approach. The advantages of using machine learning methodologies is to efficiently compare a very high amount of data, obtaining more confident results. In particular, we aligned the uc.8+ nucleotide sequence to the Ensembl transcript annotation database. The deep learning analysis identified 46 sequences spread in 45 genes, with an average length of about 19 bp (in the range 17–27). The results of the deep learning analysis for the subcellular localization of uc.8+ indicated a prevalence of lncRNA sequences located at nuclear sites (Figure 2A). In fact, 35 of these sequences, about 76% of the total number of identified 46 sequences, referred to a nuclear localization. In particular, 18 nuclear predicted sequences, about 52% of the nuclear ones, returned a prediction probability higher than 75%, supporting a nuclear localization of uc.8+. While, of the remaining 11 sequences predicting cytosolic localization, representing the 24% of the total identified 46 sequences, there are 7 sequences, about 64% of the predicted cytosolic ones, which partially overlapped to a number of the nuclear sequences. Only UUUUUUUUUUUUCUUUCUUUCUGC, AUAAAUAAAUUUAUAUC, AAAGAAUUAAUGAGUUGGUAG and UUUCCUCUGGCUUAGGU are unique cytosol predicted sequences. For the purpose of assessing the weight (importance) of cytosol predicted regions in the sequence of the lncRNA, such as the availability of these sequences for a possible interaction with specific proteins, we determined the predicted secondary structure of the whole uc.8+ (Figure 2B). The distribution of the sequences on the predicted secondary structure of uc.8+ (Figure 2C,D), as expected, showed the large extent of sequences predicting a nuclear respect to a cytoplasmic localization and the overlapping in the structure of some cytosolic and nuclear sequences (Appendix A).

### 2.3. uc.8+ Expression and Subcellular Localization in BlCa Samples and Cell Lines

BlCa tissue biopsies were further screened to assess uc.8+ subcellular localization. In Figure 3A, the general ISH profiles for biopsies from tissue samples of pts belonging to LG and HG groups are shown. In the image magnifications, the simultaneous presence of uc.8+ in the cytoplasm and nuclei of cells was prevalently observed in the samples belonging to the LG group, while a more defined cytoplasmic localization was generally observed in the samples from HG group. In detail, uc.8+ had a clear cytoplasmic localization in 23 (46%) HG samples, whereas it showed a nuclear-restricted or a cyto-nuclear localization in the remaining 19 (38%) and 8 (16%) HG samples, respectively (Figure 3B).

This result is completely inverted in the samples from LG group where the cyto-nuclear localization was observed in 10 samples (44%), while only 7 (30%) and 6 (26%) showed a nuclear- or a cytoplasm-restricted localization, respectively (Figure 3B, Appendix A). In general, we observed a higher overall uc.8+ signal intensity in the LG samples, with respect to the HG ones (Appendix A). In particular, these differences are marked in the samples with nuclear or cyto-nuclear localization. Analyzing data with respect to the intensity-score subgroups of uc.8+, we detected a cytoplasm-restricted localization in 8 biopsies out of 29 (28%) in HI-s group and in 21 biopsies out of 29 (72%) in LI-s group (Table 1), while nuclear-restricted localization was detected in 11/26 biopsies (42%) in HI-s group and 15/26 (58%) in LI-s group (Table 1). Differently, the number of biopsies showing both cytoplasmic and nuclear localization was higher in the HI-s (11 out of 18 biopsies) with respect to LI-s ones (7 out of 18 biopsies) (Table 1). These data support the assignment of the cyto-nuclear localization of uc.8+ with a less tumor malignancy.

### 2.4. uc.8+ Localization in Bladder Cancer Cell Line

To confirm these data, we also measured the subcellular localization of uc.8+ in BlCa cells by preparing nuclear and cytoplasmic fractions from urothelial carcinoma cell lines J82. As control of the cellular fractionation assay, we used GAPDH, U1 snRNA and the mitochondrially retained 12S rRNA, (Figure 3C), confirming the absence of cross contamination between the nuclear and cytoplasmic preparations. As reported in Figure 3C, uc.8+ was significantly enriched in the cytoplasmic (69%) respect to the nuclear (31%) fractions. We further separated the chromatin fraction by using the long noncoding RNA XIST, as a canonical chromatin-associated lncRNA. We found that approximately 25% of nuclear-localized uc.8+ transcripts were chromatin-enriched (Figure 3D). Additionally, we have already performed in a previous paper [14] the *in situ* hybridization experiment on J82 bladder cancer invasive cell line (T3–G3), using a modified fluorescent probe (PNA) (Appendix A). The experiments showed a clear cytoplasmic localization of uc.8+, in agreement with our ISH data from patient specimens.

### 2.5. uc.8+ Expression and Histological Localization in Tumor-Surrounding Stroma

Subsequently, we measured uc.8+ expression also in the tumor-surrounding stroma taking advantage of a tissue microarray (TMA) containing the stroma tissue of all the pts previously analyzed. The ISH was repeated twice onto different TMA slides and identical size region of interest (ROI) was selected in tumor corresponding tissue to compare paired stroma and epithelium from the same patient (Figure 4A). The analysis of uc.8+ intensity in paired tumor tissue and stroma revealed that uc.8+ was significantly lower in the stroma of LG pts compared to the paired tumor tissues, whereas in HG pts no significant differences were detected (Figure 4B,C). We observed a significant increase in uc.8+ intensity in the stroma of HG compared to LG, suggesting a possible role of uc.8+ not only in early phases of tumorigenesis inside cancer cells, but also in the tumor microenvironment and in tumor invasiveness.

In most of the stroma biopsies analyzed, we measured a cytoplasm-restricted localization for uc.8+ in about 54% of the HI-s group and 56% of the LI-s group. Whereas in about 7 and 32% of biopsies from HI-s and LI-s groups, respectively, a clear nuclear staining for uc.8+ was detected. Finally, only a small fraction (11%) of biopsies from the HI-s group and none of those from the LI-s group biopsies showed staining for uc.8+ in both intracellular compartments. Moreover, in the 28% of HI-s biopsies the stroma was not involved, and the presence of uc.8+ was not detected, also this happens in the 12% of biopsies from the LI-s group. These data indicated a prevalent cytoplasmic localization of the uc.8+ signal in tumor-surrounding stroma. The same data plotted against the BlCa stage highlighted the prevalent cytosol localization of uc.8+ in the stroma of LG and HG pts, also an increase in the number of LG pts with very low or absent uc.8+ expression, was observed (Figure 4D). These results suggest that, as described for other lncRNAs species, uc.8+ may undergo nucleus-cytoplasm shuttling, at higher extent in tumor cells rather than in the surrounding stroma.

### 2.6. Prediction of uc.8+ Interactions in Bladder Cancer

Despite the number of predicted nuclear sequences, it is interesting to observe that several sequences assigned to a cytosol localization are in single strand. In particular, the long U-repeats (6–7 nucleotides) in multibranched loop are predicted to interact with proteins involved in the transport of RNA from the nucleus to the cytoplasm, such as an ATP-dependent RNA helicase (DDX19B), which is associated with nuclear pore complex cytoplasmic fibrils and involved in mRNA export from the nucleus [16]. DDX19B functions as a remodeler of ribonucleoprotein particles, replacing the proteins bound to nuclear mRNA with the cytoplasmic mRNA binding proteins. Another protein is the heterogeneous nuclear ribonucleoprotein A1 (HNRNPA1), which may bind to specific miRNA hairpins, and is involved in several functions: packaging of pre-mRNA into hnRNP particles; transport of poly(A) mRNA from the nucleus to the cytoplasm; modulate splice site selection [17,18,19]. Moreover, this latter protein can also recognize the 75% of unique cytosol-predicted sequences present in the whole uc.8+. Additionally, the short sequence AGAC in a hairpin loop is recognized by a Nuclear RNA export factor 1 (NXF1), involved in the export of mRNA from the nucleus to the cytoplasm [20,21]. Interestingly, the expression of DDX19B and NXF1, results altered in stage I of BlCa as found in the TCGA database (Figure 5A–C), correlating with the higher uc.8+ expression.

Moreover, we found that DDX19B and NXF1 share the same network of protein-protein interactions with a high confidence value (0.884) (Figure 5D). It was suggested that in human cells DDX19B functions on the cytoplasmic side of the nuclear pore complex and is required for the final release of mRNA from NXF1 into the cytoplasm [22]. All this evidence strongly supports a possible correlation between the cytosolic transport of the nuclear uc.8+ during the tumor and/or cell life, with a cytosolic role of this lncRNA in order to fulfill its cellular functions. 

### 2.7. DDX19B and NXF1 Involvement into Nuclear Export of uc.8+

In order to support the in silico evidence of a mechanism of cytosolic transport of uc.8+ involving DDX19B and NXF1, we performed DDX19B and NXF1 silencing (Figure 5E,F), followed by RT-PCR of uc.8+ in J82 cells nuclear/cytoplasmic preparations (Figure 5G). At 48h in control cells, a substantial proportion of uc.8+ was found in the cytoplasm and nucleus (53 and 46%, respectively) (Figure 5G). In DDX19B and NXF1-silenced cells, the proportions of cytoplasmic uc.8+ were strikingly reduced to about 22 and 15%, respectively (Figure 5G). In contrast, in DDX19B and NXF1-silenced cells, the uc.8+ remained predominantly nuclear (Figure 5G).

## 3. Discussion

Since Calin et al.’s 2007 study [23], the differential expression of T-UCRs has been described in several types of cancer, and a plethora of studies has been conducted in order to characterize the role of T-UCRs in carcinogenesis [24]. To date, differential expressions of 286 (59.46%) T-UCRs (out of 481) have been associated with several types of tumor and only of 4% of T-UCRs are known molecular details [10]. Interestingly, few T-UCRs are differentially expressed in specific tumors, and this is the case uc.8+ in BlCa. 

In contrast to extensive genomic annotation of UCRs transcripts, far fewer have been characterized for subcellular localization and cell-to-cell variability. Subcellular patterns of T-UCRs provide fundamental insights into their biology. LncRNAs must localize to their site of action; thus, their location in the cell is important. Subcellular localization of lncRNA can range between nucleus and cytoplasm [25]. In particular, nuclear lncRNAs are prevalently involved in chromatin interactions, transcriptional regulation and RNA processing, while cytoplasmic lncRNAs can modulate mRNA stability or translation and influence cellular signaling cascades [25]. In our previous study, we showed that uc.8+ was located in both nuclear and cytoplasmic compartments of BlCa cell lines [14]. In the current study, we applied the ISH-RNA method for a direct observation of uc.8+ lncRNAs expression and localization in tissue sections from pts affected by bladder cancer. 

We found that uc.8+ is significantly increased in LG compared to HG pts, with a significant high expression in Ta clinical stage, suggesting an early alteration of this lncRNA in BlCa development and positioning uc.8+ as a hallmark of LG BlCa. Next, we explored the association between uc.8+ expression pattern and its subcellular distribution in bladder cancer tissues. The simultaneous presence of uc.8+ in the cytoplasm and nuclei of cells, was prevalently observed in the sample belonging to the LG group. This ubiquitous subcellular localization of uc.8+ transcripts may contribute to the function of uc.8+ both in nucleus and cytosol. Cytoplasmic localization was, conversely, preferentially observed in samples from the HG group. To sum up, these results suggested the assignment of the cyto-nuclear localization of uc.8+ to a less tumor malignancy and helped to better understand that the target localization diversity of lncRNAs is an important feature closely related to clinical prognosis.

Furthermore, the analysis of tumor-surrounding stroma showed a significant increase in uc.8+ ISH signal intensity in the stroma of HG compared to LG, suggesting a possible role of uc.8+ not only in cancer cells in the early phases of tumorigenesis but also in tumor microenvironment and tumor invasiveness. In addition, in tumor-surrounding stroma, the uc.8+ signal showed a prevalent cytoplasmic localization, suggesting that uc.8+ might undergo nuclear-cytoplasmic shuttling at higher extent in tumor cells rather than in the surrounding stroma.

The predominant nuclear localization of the uc.8+ does not exclude its functions in the cytosolic compartment, considering that possible target-proteins could be functioning in different compartments and could have multi localization in the cell. Despite the high number of sequences predicted by deep-machine-learning approach, locating uc.8+ predominantly at nuclear site, the presence of sequences that are recognized by RNA carrier proteins, for which a dependence of the expression on the tumor stage was observed, supports the displacement of uc.8+ in other subcellular compartments. In general, very few is still known about the displacement nucleus-cytosol mechanism for lncRNA, but, considering the cell energy saving principle, it is logical to hypothesize that the same proteins active in the mRNA pathways could be used also by lncRNAs.

Instead, the differences in uc.8+ subcellular localization, being the target localization of a lncRNA indicative of its functions, is harder to interpret. According to our model (Figure 6) in which uc.8+ is transcribed in the nucleus, displaced in the cytosol and after degraded, carrying out its function in both compartments, the Ta clinical stage could be related to a defect in the feedback of transcription control or in the lncRNA degradation, with the consequent massive increase in the amount of uc.8+ on both sides.

Differently, the exclusive subcellular localization of uc.8+ in the cytosol represents a more dramatic event, because involving the entire pathway of displacement and affecting the nuclear function of the lncRNA. In fact, the amount of transcribed uc.8+ is still increased in HG pts, but is almost absent in the nucleus, therefore, all the nuclear uc.8+ functions are cleared. These events resulted in more aggressive/invasive cancer.

The assignment of the cyto-nuclear localization of uc.8+ with a less tumor malignancy suggested that in high grade tumors more than expression, cellular localization is relevant.

Our study of uc8+ in bladder cancer produced the foundations for its implementation into clinical practice. External validation of this uc8+-based classifications and scores is now required and its clinical impact will be measured according to the results of future prospective trials. However, certain limitations regarding the use of uc8+ molecules should be considered. Subcellular localization still requires the biopsy specimen, an invasive technique for the patient, and some precautions must be considered to counteract the sample heterogeneity, which is a hot issue in diagnostic histopathology. Nevertheless, molecular analysis carried out by the subcellular localization of uc.8+ could better identify subtypes of bladder cancer and give a more precise patient stratification toward the therapies. Therefore, uc8+-based biomarker may have a good feasibility in clinical practice.

## 4. Materials and Methods

### 4.1. Clinical Characteristics of Bladder Cancer Patients

A total of seventy-three pts were accrued, with a median age at diagnosis of bladder cancer of 68 years, ranging from 44 to 89 years (Table 1). There was no significant difference in age between male and female pts. Table 1 presents the clinical and histopathological characteristics for the 73 study pts. The majority of pts were male (84%; *n* = 61). A total of sixty-eight percent of pts (*n* = 50) were diagnosed with High Grade (HG) and 32% (*n* = 23) with Low Grade (LG) BlCa. Within the HG group, only 4% of pts (*n* = 2) were diagnosed at superficial stage of the disease (Ta), while 58% (*n* = 29) were diagnosed at stage T1–T2, 38% (*n* = 19) were diagnosed at stage T3–T4. On the contrary, within the LG group, 61% of pts (*n* = 14) were diagnosed at superficial stage of the disease (Ta), while 35% (*n* = 8) were diagnosed at stage T1–T2 and only 4% (*n* = 1) were diagnosed at stage T3–T4. Nineteen percent (*n* = 14) of pts presented metastasis, all of belonging to the HG group. The majority of pts presented tumor infiltration in the subepithelial connective tissue lamina propria (66%; *n* = 49), while 16% (*n* = 11) and 18% (*n* = 13) showed infiltration in muscularis propria bladder wall and perivesical tissue, respectively. A total of twenty-two pts diagnosed with HG BlCa were exposed to chemotherapy, 55% (*n* = 12) were treated with neoadjuvant therapy and 45% (*n* = 10) with adjuvant therapy.

### 4.2. Ethics Statement

The samples were collected at “Biobanca Istituzionale dei Tessuti Istituto tumori Pascale”, Naples, Italy as approved by “Istituto Nazionale Tumori di Napoli”, IRCCS “G. Pascale” in the Resolution of the Extraordinary Commissioner; number: 15, date: 20 January 2016. We declare that informed consent for the scientific use of biologic material was obtained from all pts.

### 4.3. Tissue Microarray Building

A total of 73 tissue samples selected from 2004 to 2016, at the National Cancer Institute “Giovanni Pascale″ of Naples, were used for building a prognostic tissue microarray (TMA). We included only samples with at least 2 representative tumor paraffin embedded tissue. Tumor tissues included 23 Non-Muscle Invasive bladder transitional cancer” (NMIBC) and 50 Invasive tumors infiltrate muscularis propria: “Muscle-Invasive bladder transitional cancer” (MIBC). All tumors and controls have been reviewed by two experienced pathologists according to WHO/ISUP 2016. Sections of 4 µm were obtained from each block and stained with hematoxylin and eosin. Two different areas, rich in non-necrotic tumoral cells were identified on corresponding haematoxylin and eosin-stained whole slides and marked. Whenever possible, one normal tissue was identified. The surgical donor sample was punched in a precise area and a tissue cylinder with a diameter of 1 mm was transferred to the recipient paraffin block using a semiautomated tissue array (Galileo TMA). We used two different tissue cylinders 1.0 mm diameter for TMA construction as suggested by Eskaros AR et al to overcome heterogeneity of bladder cancer. All the donor cores were formatted into one recipient block. H&E staining of a 4-µm TMA section was used to verify all samples.

### 4.4. In Situ RNA Hybridization on Paraffin Sections

The uc.8+ RNA antisense probe was synthesized from 1ug of linearized plasmid in presence of digoxigenin-11-UTP using DIG RNA labelling kit (Roche, Mannheim, Germany, cat#11175025910) following the manufacturer instructions, cat#. Paraffin embedded tissue sections were dewaxed in xylene, rehydrated, postfixed in 4% paraformaldehyde, digested with proteinase K (10 µg/mL) for 10 min, subjected to acetylation with 0.1 M triethanolamine—0.25% acetic anhydride for 15 min, pre-hybridized for 1 h, and then hybridized at 65 °C overnight in 50% formamide; 0.25% sodium dodecyl sulfate; 10% dextran sulfate; 1× Denhardt solution; Tris HCl (pH 7,5, 10 mM); NaCl (600 mM); EDTA (1 mM); transfer RNA (200 g/mL); salmon sperm DNA (100 g/mL), using a probe concentration of 0.8 μg/mL. After hybridization, tissue sections were washed in 1× saline sodium citrate (SSC) buffer-50% formamide solution at 65 °C for 30 min and then in 2× SSC buffer for 20 min and in 0.2× SSC twice for 20 min each. After the washes, tissue sections were incubated overnight at 4 °C with alkaline-phosphatase-conjugated anti-digoxigenin antibodies (1:2000; Roche). After seven washes in maleic acid buffer containing 0.1% Tween 20 (MABT) for 1 h each and three washes in 0.1 M NaCl—0.1 M Tris HCl ph 9.5—0.05 M MgCl2—0.1% Tween 20 (NTMT) solution for 10 min each, tissues sections were incubated with nitro blue tetrazolium chloride (NBT) and 5-bromo-4-chloro-3-indolyl-phosphate (BCIP), 4-toluidine salt solution (Roche) and developed a blue color. The reaction was checked under microscope and blocked with several washes in PBS at the desired time point. Tissue sections were then counterstained with eosin (Sigma-Aldrich, St. Louis, MO, USA), to evaluate cytoplasmic uc.8+ expression, or with fast red solution (Sigma-Aldrich), to evaluate its nuclear amount, dehydrated and mounted in Eukitt mounting reagent. Stained sections were examined and photographed using a Leica MZ12 dissection microscope and a Nikon ECLIPSE Ni microscope (Nikon, Tokyo, Japan).

### 4.5. Image Processing and Analysis

Image processing and analysis were performed using ImageJ software [26]. In detail, a macro was designed to process all the images in a rigorous and standardized manner consisting in converting the images in the grayscale format (8-bit format) and determining the threshold with the Maximal Entropy algorithm to segment and quantify the signal intensity.

The output used for downstream analysis was the Mean Gray Intensity (MGI) representing the level of uc.8+ expression. A negative control (sample without probe) was used to normalize the MGI computed to obtain a value we named Intensity Score. All the results obtained were compared with the scoring performed independently by three pathologists and were considered compatible.

### 4.6. Transfection of siRNAs against uc.8+ in BlCa Cells

In total, two siRNA duplexes targeting different reading frames of DDX19B and NXF1 (Dharmacon, Lafayette, CO, USA) were used for RNAi-mediated knockdown. ON-TARGETplus non-targeting siRNA (Thermo Scientific, Waltham, MA, USA) was used for the siRNA control. The siRNA transfections were done using INTERFERin (Polyplus-transfection) at 10 nM. Depletion was verified by preparing J82 total cell lysates 48 apost-RNAi treatment and analyzing by qRT-PCR.

### 4.7. Cellular Fractionation

A total of 1 × 10^7^ J82 cells was washed twice with cold PBS and then were resuspended and incubated in hypotonic buffer (50 mM HEPES, pH 7.5, 10 mM KCl, 350 mM sucrose, 1 mM EDTA, 1 mM DTT and 0.1% Triton X-100) on ice for 10 min. The suspension was centrifuged 5 min at 2000 g, and the cytoplasmic fraction was collected in the supernatant. The nuclear pellets, after additional washing, was resuspended in lysis buffer (10 mM HEPES, pH 7.0, 100 mM KCl, 5 mM MgCl2, 0.5% NP-40, 10 μM DTT and 1 mM PMSF), in order to prepare the nuclear lysate.

### 4.8. Nuclear Fractionation

To isolate the chromatin-enriched RNA, the chromatin pellets and the soluble nucleoplasm were prepared from the nuclear extract as described [27]. Nuclear fractionation was performed similar to [28,29]; 10–20 × 10^6^ adherent J82 cells were cultured in Dulbecco’s modified Eagle’s medium and supplemented with 10% fetal bovine serum under a humidified atmosphere of 95% air and 5% CO^2^ at 37 °C to ~80% confluence, washed in 1xPBS, and then recovered by scraping and centrifugation (500× *g*, 5 min, 4 °C). For transcriptional inhibition experiments, 5,6-dichloro-1-β-Dribofuranosylbenzimidazole (DRB) (Sigma Aldrich) was dissolved in DMSO to 75 mM, then added to cells for two hours at 100 μM final concentration. An equal volume of only DMSO was added to other cells as a mock control. Cell pellets were resuspended in 2.5 × volumes of buffer A (10 mM HEPES pH 7.5, 10 mM KCl, 10% glycerol, 340 mM sucrose, 4 mM MgCl_2_, 1 mM DTT, 1 × Protease Inhibitor Cocktail (1 mM PMSF, 1 mM ABESF, 0.8 μM aprotinin, 20 μM leupeptin, 15 μM pepstatin A, 40 μM bestatin, 15 μM E-64)), and then an equal volume of buffer A with 0.2% (*v*/*v*) Triton X-100 was added and the mixture incubated on ice for 12 min to lyse cells, followed by centrifugation (1200× *g*, 5 min, 4 °C). The crude nuclear pellet was resuspended in 250 μL NRB (20 mM HEPES pH 7.5, 50% Glycerol, 75 mM NaCl, 1 mM DTT, 1 × protease inhibitor cocktail), transferred to a microcentrifuge tube and centrifuged (500× *g*, 5 min, 4 °C) to wash. The pellet was resuspended in 250 μL NRB, and then an equal volume of NUN buffer (20 mM HEPES, 300 mM NaCl, 1M Urea, 1% NP-40 Substitute, 10 mM MgCl2, 1 mM DTT) was added and incubated 5 min on ice, then centrifuged (1200 × *g*, 5 min, 4 °C). The soluble nuclear extract supernatant was transferred to another tube, and the depleted nuclear pellet was resuspended in 1 mL buffer A to wash, transferred to another microcentrifuge tube, and centrifuged (1200× *g*, 5 min, 4 °C). Resulting purified chromatin pellets were resuspended in 50 μL buffer A. TRIzol (0.5 mL) was added to the re-suspended chromatin RNA, and to 20% (*v*/*v*) of nuclear-soluble extracts for RNA extraction.

The manufacturer’s protocol was followed to obtain an aqueous RNA layer, which was used as input for RNA CleanConcentrator™-25 columns (Zymo Research, Irvine, CA, USA). In-tube DNase digestion was performed according to the manufacturer’s protocol, and pure chromatin and nuclear-soluble extract RNA fractions were eluted in 50 μL RNase/DNase-free water.

### 4.9. Gene Expression Analysis

After isolation of RNA from whole cell lysate or specific subcellular fractions by using TRIzol reagent (Invitrogen, Waltham, MA, USA), the RNA levels for a specific gene were measured by qRT-PCR (starting with 50–100 ng RNA sample per reaction) using Real-Time PCR Detection System (Bio-Rad, Hercules, CA, USA) according to the manufacturer’s instructions. The data obtained from qRT-PCR were normalized to mitochondrial-retained 12S rRNA, nuclear-localized U1 snRNA, or chromatin-associated XIST RNA. Primers used in this study for qRT-PCR are:XIST-F—5′TGATCCCATTGAAGATACCACGCTG3′;XIST-R—5′TGGCAACCCATCCAAGTAGATTAGC3′;rRNA-F—5′ACTGCTCGCCAGAACACTACGA3′;rRNA-R—5′GTCTTTACGTGGGTACTTGCGCT3′;snRNA-F—5′TCCCAGGGCGAGGCTTATCCATTG3′;snRNA-R—5′GCGAACGCAGTCCCCCACTACCAC3′;GAPDH-F—5′GTCAAGGCTGAGAACGGGAAGCT3′;GAPDH-R—5′GCCTTCTCCATGGTGGTGAAGA3′;uc.8+-F—5′-GGTCGCCATGGATATGACA3′;uc.8+-R—5′CACTGTGGCTTTAAACTCAGGA3′;Ddx19B-F—5′CGTCCATCCAAGATACAAGAGA3′;Ddx19B-R—5′ TTGGGCAATTAAGTTCTGTGG3′;NXF1-F—5′ATTTGGATCCATGGCGGACGAGGGGAAGT3′;NXF1-R—5′AATATGCGGCCGCTCACTTCATGAATGCCACTTCTGG3′.

### 4.10. LncRNA Subcellular Localization with Deep Learning

In this work, a Deep Learning-based approach was used for locating lncRNAs contained in a DNA sequence and for estimating its percentage of presence both in the cytosol and in the nucleus. In particular, we adopted DeepLncRNA [15], which is a feed-forward multilayer Deep Neural Network [30]. DeepLncRNA is learned on a dataset collected by starting from paired-end strand-specific RNA-sequencing data from human cell lines from the ENCODE project [15]. The architecture consists of one input layer and three hidden layers. Hidden layers use the Rectified Linear Unit activation function and the output layer a softmax activation function. Input layer consists of 1582 neurons, and the hidden layers of 32-16-8 neurons, respectively. For the hidden layers dropout was used for reducing the overfitting. Input dropout was also considered for generalization of the model. Moreover, regularization was applied using the L1 and L2 weight penalties to the cost function. The model was trained with stochastic gradient descent using the backpropagation algorithm. A cross-validation mechanism for making generalization of the model was also used. Data were divided in 70% for the training set and 15% for both validation and test sets.

### 4.11. In Silico Data Analysis of Gene Expression and Protein-Ligand Interactions

The prediction of proteins binding the identified cytosol predicted sequences was carried out using the ATTRACT database [31] build on experimentally validated RNA binding proteins and associated motifs. The searches were restricted to the range from 4 to 12 nucleotides as minimum and maximum length of the motif, respectively, selecting *Homo sapiens* as organism. The search results recognizing multiple motifs with maximum length, and showing the best quality score in terms of affinity between RNA and binding site, have been considered.

Prediction of the uc.8+ secondary structure was carried out on the whole 2435-bp length transcript sequence, using the RNAfold software (Vienna RNA Package) [32,33].

Gene expression data of bladder cancer were downloaded from Genomics Data Common Data Portal (GDC Data Portal; https://portal.gdc.cancer.gov/) [34]. The follow up clinical information was available for 412 pts with bladder cancer from TCGA-BLCA project, Gene Expression Quantification data type, RNA-Seq experimental strategy, Transcriptome Profiling data category, and HT-Seq Counts workflow type. The data were downloaded, pre-processed and analyzed in the R environment, by means of libraries available in Bioconductor online repository [35], such as TCGAbiolinks [36], Summarized Experiment [37], edgeR [38], and ggpubr [39]. 412 pts with bladder cancer were divided into 4 tumour stages: 4 stage I, 130 stage II, 142 stage III, and 136 stage IV. 19 control cases were extracted from the healthy tissue of the same pts. Data counts were normalized in counts per million (CPM) and log2-transformed for a better representation in terms of distribution for expression level data. The Kruskal–Wallis test was performed in order to provide the statistical significance in the difference between the control cases and each stage of the data.

The network of protein–protein interaction was searched for each of the predicted proteins, which could bind uc.8+, using the STRING database [40]. The confidence mode was used to determine the degree of confidence for the prediction of the interaction between these proteins. The network is described with nodes which are the proteins, and the edges representing the predicted associations (line thickness indicates the strength of data support).

### 4.12. Statistical Analysis

All statistical analyses were performed using the R Studio software package Team (2015) (RStudio: Integrated Development for R, RStudio, Inc., Boston, MA, USA, URL http://www.rstudio.com/). Shapiro–Wilk’s test was performed to the assessment of normality of the distributions and Bartlett’s test-to-test equal variance. Associations between clinicopathological parameters and uc.8+ expression were analyzed using Wilcoxon rank sum test and Welch’s *t*-test in the case of two or more groups had to be compared respectively. Significant variables in univariate models were further analyzed by multiple logistic regression model using the tumor grade as response and age, sex, and uc.8+intesinty as explanatory variables. We considered the regression coefficients with a *p*-value < 0.05 statistically significant. Overall survival was calculated and survival curves were plotted using the Kaplan–Meier method. Differences between groups were compared using log-rank tests. All tests were two- sided and *p* values < 0.05 were considered statistically significant

## 5. Conclusions

The subcellular localization of lncRNAs could be a key feature to understanding their role in cancer. The analysis of uc.8+ in bladder cancer tissues, linked its accumulation and subcellular distribution to clinical phenotype.

## Figures and Tables

**Figure 1 cancers-13-00681-f001:**
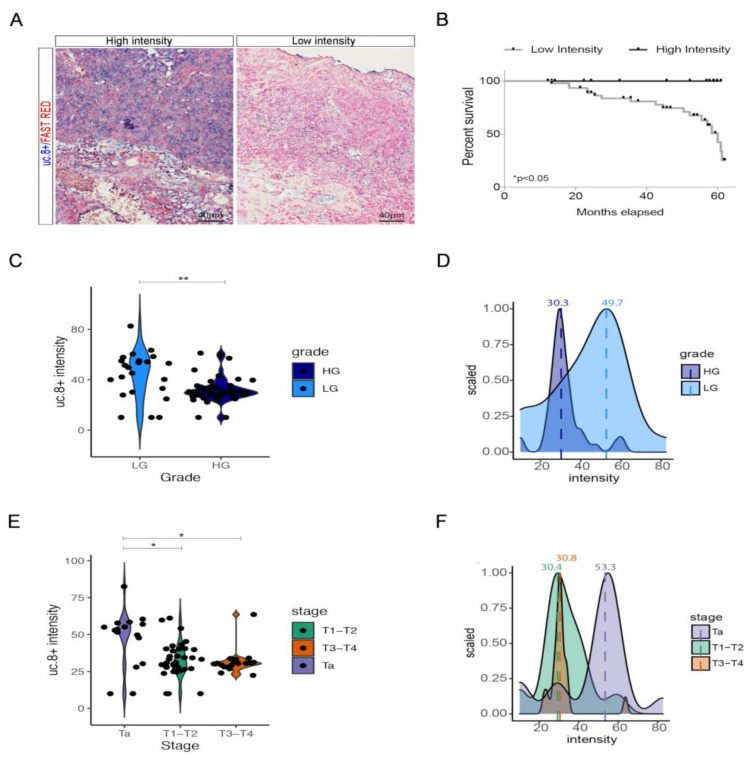
(**A**) Representative pictures of RNA in situ hybridization of uc.8+ in BlCa pts. (**B**) Survival plot of pts with low and high uc.8+ intensity. (**C**) Violin plot showing uc.8+ intensity score values across tumor grade and (**D**) frequency plot showing the scaled distribution of uc.8+ intensity value according to the patient grade and their relative median values (high grade = 30.3; low grade = 49.7). (**E**) Violin plot showing uc.8+ intensity score values across tumor stage and (**F**) frequency plot showing the scaled distribution of patient uc.8+ intensity value according to the stage and their relative median values (Ta = 53.3; T1–2 = 30.4; T3–4 = 30.8). * *p*-values < 0.05, ** *p*-values < 0.01.

**Figure 2 cancers-13-00681-f002:**
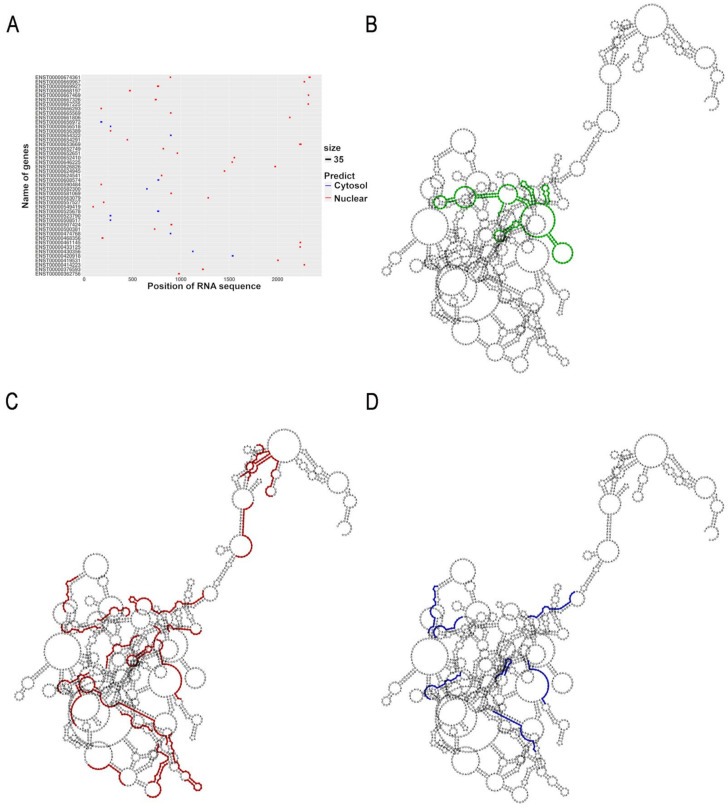
(**A**) Plot of the distribution in the uc.8+ sequence of the predictive genes for the nuclear and cytosolic localization by deep machine learning analysis. (**B**) Predicted secondary structure representation of the whole uc.8+, in green the ultraconserved region. (**C**) Sequences associated with a nuclear localization (in red). (**D**) Sequences associated with a cytosolic localization (in blue).

**Figure 3 cancers-13-00681-f003:**
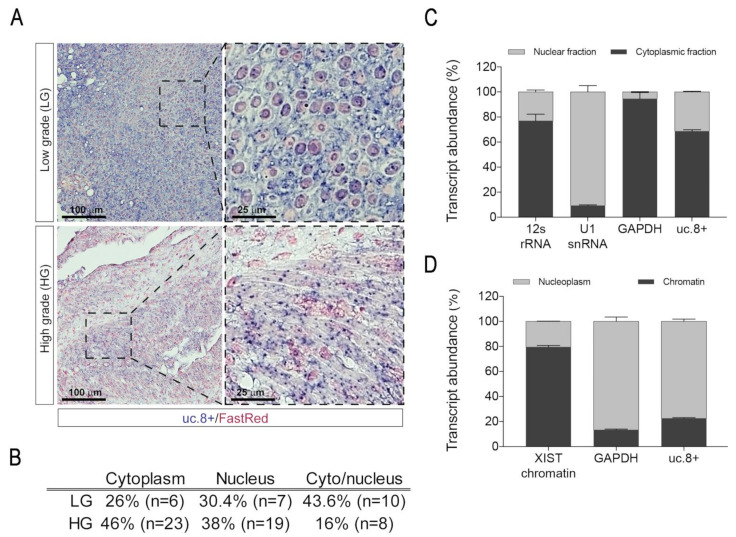
(**A**) Representative ISH images for uc.8+ sub-cellular localization in LG and HG BlCa pts. On the right, magnification of the squared areas are reported. (**B**) Table showing the uc.8+ sub-cellular distribution in samples belonging to the HG and LG groups. (**C**) qRT-PCR assay following nuclear/cytoplasmic fractionation detecting the distribution of the uc.8+ in J82 cells. (**D**) qRT-PCR assays detecting the distribution of the uc.8+ in chromatin and nucleoplasm extract from J82 cells. The qRT-PCR data are presented as means ± SD of percentage of relative transcript abundance from three independent experiments performed in triplicate. XIST, were assessed as chromatin-associated lncRNA, and GAPDH mRNA, were assessed as chromatin fractionation controls.

**Figure 4 cancers-13-00681-f004:**
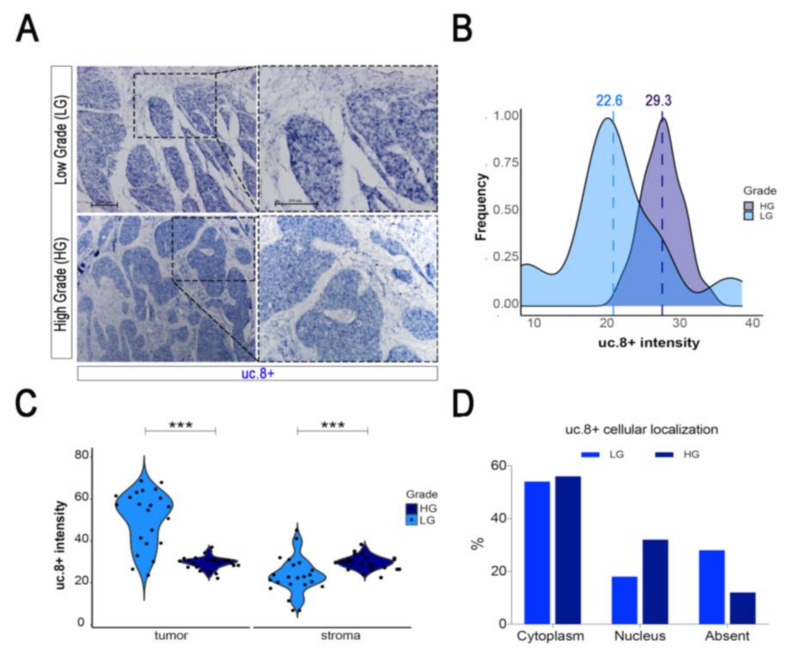
(**A**) Representative pictures of RNA in situ hybridization of uc. 8+ in BlCa pts-surrounding stroma. (**B**) Frequency plot showing the scaled distribution of uc.8+ intensity in bladder cancer-surrounding stroma according to the grade. (**C**) Violin plots showing uc.8+ intensity score values in tumor and surrounding stroma across tumor grade. (**D**) Bar plots showing uc.8+ cellular localization according to the grade stratification. *** *p*-values < 0.001.

**Figure 5 cancers-13-00681-f005:**
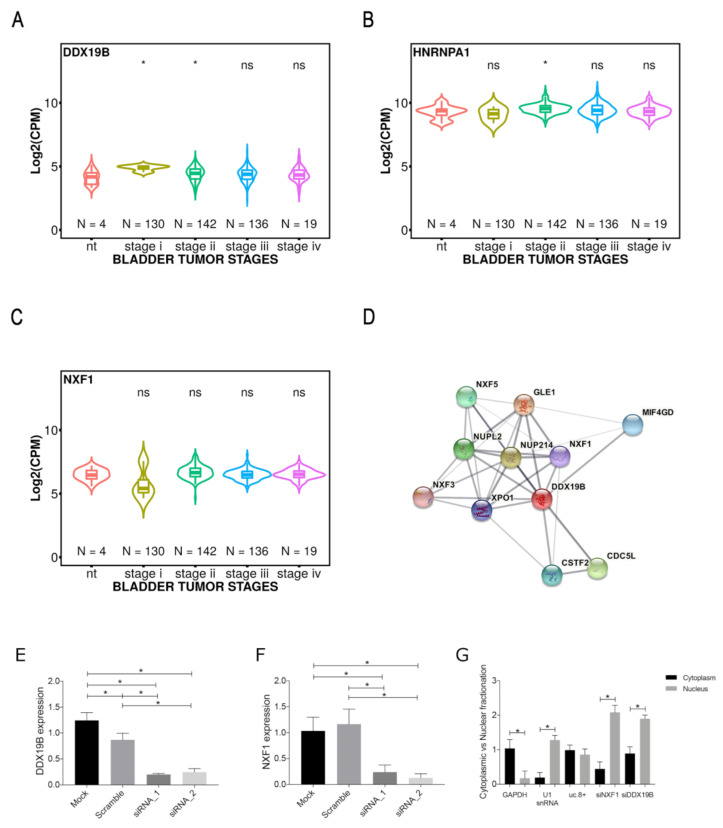
Violin plot for (**A**) DDX19B, (**B**) NXF1 and (**C**) HNRNPA1 expressions (CPM) in TCGA-BLCA project, at different tumor stages. (**D**) network of known and predicted protein-protein interactions for DDX19B and NXF1 as described in the STRING database, line thickness indicates the strength of data support. (**E**) J82 cells were transfected with siRNA_1 and siRNA_2 anti-DDX19B or siRNA scramble. (**F**) J82 cells were transfected with siRNA anti-NXF1_1 and siRNA_2 or siRNA scramble. The DDX19B and NXF1 level was determined by qRT-PCR. (**G**) At 48 h post-transfection with siRNAs targeting DDX19B (siRNA1 + 2) and NXF1 (siRNA1 + 2) the level of uc.8+ was determined by RT-qPCR and expressed as fold change vs. control. Data are expressed as the mean ± SD of triplicate values. * *p*-values < 0.05.

**Figure 6 cancers-13-00681-f006:**
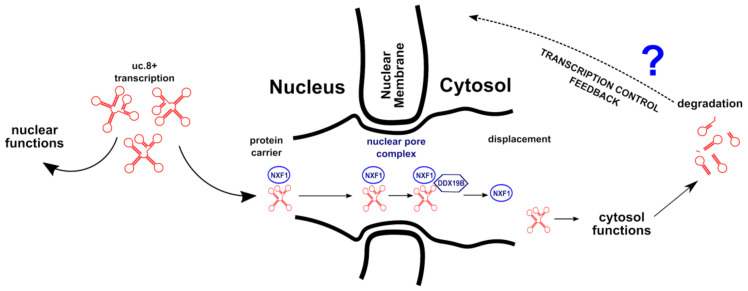
Hypothesized mechanisms of uc.8+ transcription and regulation.

**Table 1 cancers-13-00681-t001:** Correlation between uc.8+ expression and clinicopathologic features of bladder cancer pts.

	All Patients	Intensity-Score Subgroups			
	*n* = 73	Low Intensity Score (<35) *n* = 46	High Intensity Score (≥35) *n* = 27	Chi-Square	*p*-Value
Age at diagnosis				0.440	0.735
<60 years	17	11	6		
≥60 years	56	35	21		

Sex				0.006	0.936
Male	61	37	24		
Female	12	9	3		

Grade				16.394	5.14 × 10^−5^
High	50	39	11		
Low	23	7	16		

Stage				18.34	1.04 × 10^−4^
Ta	16	4	12		
T1–T2	37	24	13		
T3–T4	20	18	2		

Metastasis				8.967	2.75 × 10^−3^
absent	59	33	26		
present	14	13	1		

Therapy				10.084	0.007
no	51	27	24		
yes	22	19	3		

Localization				3.582	0.058
cytoplasm	29	21	8		
nucleus	26	15	11		
nucleus-cytoplasm	18	7	11		

All pts were classified according to the 1997 UICC TNM classification for the stage and OMS 2004 for the grade.

## Data Availability

The data presented in this study are available on request from the corresponding author.

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
