# Peer review of "Subcellular Localization of uc.8+ as a Prognostic Biomarker in Bladder Cancer Tissue"

_cancers, 2021, doi:10.3390/cancers13040681_

Round 1

Reviewer 1 Report

The manuscript by Terreri et al. provides a systematical evaluation of ultra38 conserved-transcript-8+ (uc.8+) expression in biopsies and also investigate its intracellular localization. This study suggests that uc.8+ localization can be used as a prognostic biomarker for bladder cancer. Specific comments are as follows:

  1. On page 6, how do the authors conclude that “In fact, about 76% of these sequences referred to a nuclear localization, in particular 52% with a prediction probability higher than 75%” and “About 64% of the remaining number of sequences predicting cytosolic localization (24%), overlapped (even if not completely) to the nuclear ones”? How to calculate the percentage?

  1. In Figure 5, the figure legend “Violin plot for DDX19B (A), HNRNPA1 (B) and NXF1 (C) expressions” doesn’t match to the figure “NXF1 in (B) and HNRNPA1 in (C).

  1. On page 11, Please explain more clearly about the sentence “Interesting the expression of all these proteins, in particular for DDX19B and NXF1, results altered in stage I of BlCa as found in the TCGA database (Figure 5A-C), correlating with the higher uc.8+ expression”. What does “Interesting the expression of all these proteins” mean?

  1. In Figure 5E, please design some experiments to show the directly binding of NXF1 to uc.8+ in the nucleus and DDX19B to uc.8+ in the cytosol. Please also label more clearly in the Figure 5E. Are all the yellow protein NXF1 in Figure 5E? What are the green and blue proteins in Figure 5E? What is the double helix structure binding with uc.8+ shown in nuclear functions?

  1. In the materials and methods, the information about statistical analyses should be provided.

Reviewer 2 Report

This study aim is to systematically evaluate the expression of ultra38 conserved-transcript-8+ (uc.8+) in biopsies and assess its intracellular localization. They found out correlated uc.8+ levels with clinical parameters and patient survival. There were analysis indicates that uc.8+ can localize both in the cytoplasm and nucleus of bladder cells at early stage of tumorigenesis, while in tumors at advanced stages uc.8+ has a prevalent cytoplasmic localization. These data provide relevant information about uc.8+ localization as hallmark of tumor stage. Finally, using advanced computer-based techniques we predicted the binding of uc.8+ to RNA-binding proteins. Our study overall suggests that uc.8+ localization can be used as a prognostic biomarker for bladder cancer. This study , detailed, innovative, but, there are some questions need to explain : 1. Please provide IRB certificate. 2. All tissue samples were scored and categorized computing uc.8+ expression level by using the Mean Gray Intensity (MGI) score, taking advantage of ImageJ software as described in materials and methods. But the lack of normal control group, Please add the normal control data. 3. Univariate analyses showed that increased expression of uc.8+ associates with the prognosis of BlCa pts for 5-year overall survival rates (p

Reviewer 3 Report

  1. In the abstract, the authors note that determining whether uc.8+ could be used as a biomarker is a key focus – with this in minds, I think details regarding other biomarkers for bladder cancer/their accuracy should be noted in the Introduction with comment on why we need more biomarkers.
  2. Lines 83 – 85: Please include a citation(s) for this statement.
  3. Table 1: The terms ‘low’ and ‘high’ need to defined with the table (per the text ‘low’ is ‘low intensity score’ and high is ‘high intensity score’ – this needs to be stated in the table.
  4. Materials and methods section 4.1/4.3: Please state how the patient samples were selected, e.g. was this a convenience sample? Also, how were regions selected for inclusion in the tissue microarray?
  5. Line 150: Please state the hazard ratio (or odds ratio) associated with the p value.
  6. Figure 3A: Please provide a higher magnification image to help the reader see the localization.
  7. Line 228/subcellular fractionation: More details need to be provided to convince readers of the nuclear/cytoplasmic localization of uc.8+; levels of the nuclear control (U1 snRNA) and the cytoplasmic control (GAPDH) need to be assessed in both the nuclear and cytoplasmic fraction to demonstrate that there was not cross contamination between the nuclear/cytoplasmic preps. I would also like to see ISH confocal analysis of J82; this will help further determine the cellular location of uc.8+. I think this is very important to do as it the location of uc.8+ is a key part of this manuscript.
  8. The authors should calculate r values and report these to help readers understand if uc.8+ has potential utility as a biomarker.
  9. This journal has a broad readership; more time needs to be spent explaining the purpose of the studies described in section 2.2. Having a more descriptive title for this section might help, for example ‘In silico analysis indicate that uc.8+ has both nuclear and cytoplasmic binding partners’.

Reviewer 4 Report

The manuscript by Terreri et al investigates the use of a lncRNA uc.8+ asa prognostic biomaker in bladder cancer. This is a novel manuscript that looks at a poorly understood class of lncRNA, the ultra-conserved regions. Based on previous published results the authors use ISH to look at sub-cellular localisation of uc.8+ in a TMA of 73 bladder cancer samples. This is a well written manuscript that provides novel insight into the potential biomarker potential of UCR lncRNAs. That said there are a number of improvements that should be addressed before publication. 

Primarily the use of TMAs as opposed to whole sections, specifically that it should be shown that TMAs are equivalent and that the authors have accounted for possible intratumoural heterogeneity of uc.8+ expression in the samples.

Furthermore it would be useful to have some illustration of uc.8+ expression in non-malignant tissue and whether LG or HG (or other variant) of sub-cellular localisation occurs in normal tissue.

It would also be useful if the authors could provide multivariate analysis to determine whether or not prognostic significance is an independent variable or a surrogate for grade.

Finally the authors should discuss the practicality (or not) of these techniques in the clinic. 

Round 2

Reviewer 1 Report

Authors have answered most of the questions.

There is a typo of “NFX1” in the figure legend 5E-G and “DDX19” should be changed to “DDX19B” in the whole manuscript.

Author Response

In agreement with the reviewer comments, we have corrected the typo of “NFX1” in the figure legend 5E-G and changed “DDX19”  to “DDX19B” in the whole manuscript.

Reviewer 3 Report

Thank you for addressing my concerns.

Author Response

We thanks the reviewer for its comments, which permit to improve the manuscript.

Reviewer 4 Report

The authors have responded adequately to the my comments. I recommend the manuscript is accepted for publication in its present form.

Author Response

(The authors gave the same response as above.)
